# Charged Particle and Conventional Radiotherapy: Current Implications as Partner for Immunotherapy

**DOI:** 10.3390/cancers13061468

**Published:** 2021-03-23

**Authors:** Damiënne Marcus, Relinde I. Y. Lieverse, Carmen Klein, Amir Abdollahi, Philippe Lambin, Ludwig J. Dubois, Ala Yaromina

**Affiliations:** 1The M-Lab, Department of Precision Medicine, GROW–School for Oncology and Developmental Biology, Maastricht University, Universiteitssingel 40, 6229 ER Maastricht, The Netherlands; d.marcus@maastrichtuniversity.nl (D.M.); relinde.lieverse@maastrichtuniversity.nl (R.I.Y.L.); philippe.lambin@maastrichtuniversity.nl (P.L.); ludwig.dubois@maastrichtuniversity.nl (L.J.D.); 2German Cancer Consortium (DKTK) Core-Center Heidelberg, National Center for Tumor Diseases (NCT), Clinical Cooperation Unit Translational Radiation Oncology, Heidelberg University Hospital (UKHD) and German Cancer Research Center (DKFZ), Im Neuenheimer Feld 460, 69120 Heidelberg, Germany; carmen.klein@dkfz-heidelberg.de (C.K.); a.amir@dkfz-heidelberg.de (A.A.); 3Heidelberg Ion-Beam Therapy Center (HIT), Division of Molecular and Translational Radiation Oncology, Heidelberg Faculty of Medicine (MFHD) and Heidelberg University Hospital (UKHD), Im Neuenheimer Feld 450, 69120 Heidelberg, Germany; 4National Center for Radiation Oncology (NCRO), Heidelberg Institute of Radiation Oncology (HIRO), Heidelberg University and German Cancer Research Center (DKFZ), Im Neuenheimer Feld 222, 69120 Heidelberg, Germany

**Keywords:** radiotherapy, charged particle radiation, immunotherapy, immunogenicity, carbon ion, proton, clinical trials

## Abstract

**Simple Summary:**

Immunotherapy provides the unprecedented opportunity to prolong the survival of cancer patients and even cure patients with previously untreatable malignancies. Preclinical and clinical studies show that standard photon-based radiotherapy and immunotherapy can synergize in order to promote both local and systemic anti-tumor immunity and that there is still ample room for improvement. Charged particle radiation is thought to have greater immunogenic potential compared to photon radiotherapy due to more lethal unrepaired damage, higher ionization density and thus more complex clustered DNA lesions. In this review, several factors determining the success of radiotherapy combined with immunotherapies, such as composition of the tumor, radiotherapy scheme and schedule, radiation dose, the type of radiation, are addressed. Furthermore, the theoretical basis, first pieces of evidences and new insights supporting a favorable immunogenicity profile of charged particle radiation are examined, including a depiction of best of knowledge for the immune-related responses triggered by charged particles and prospective clinical trials.

**Abstract:**

Radiotherapy (RT) has been shown to interfere with inflammatory signals and to enhance tumor immunogenicity via, e.g., immunogenic cell death, thereby potentially augmenting the therapeutic efficacy of immunotherapy. Conventional RT consists predominantly of high energy photon beams. Hypofractionated RT regimens administered, e.g., by stereotactic body radiation therapy (SBRT), are increasingly investigated in combination with cancer immunotherapy within clinical trials. Despite intensive preclinical studies, the optimal dose per fraction and dose schemes for elaboration of RT induced immunogenic potential remain inconclusive. Compared to the scenario of combined immune checkpoint inhibition (ICI) and RT, multimodal therapies utilizing other immunotherapy principles such as adoptive transfer of immune cells, vaccination strategies, targeted immune-cytokines and agonists are underrepresented in both preclinical and clinical settings. Despite the clinical success of ICI and RT combination, e.g., prolonging overall survival in locally advanced lung cancer, curative outcomes are still not achieved for most cancer entities studied. Charged particle RT (PRT) has gained interest as it may enhance tumor immunogenicity compared to conventional RT due to its unique biological and physical properties. However, whether PRT in combination with immune therapy will elicit superior antitumor effects both locally and systemically needs to be further investigated. In this review, the immunological effects of RT in the tumor microenvironment are summarized to understand their implications for immunotherapy combinations. Attention will be given to the various immunotherapeutic interventions that have been co-administered with RT so far. Furthermore, the theoretical basis and first evidences supporting a favorable immunogenicity profile of PRT will be examined.

## 1. Introduction

Despite technological advances in the precise delivery of radiation that enable higher radiation doses per fraction and at the same time better sparing of surrounding normal tissues, many patients (~60%) still experience tumor recurrences after treatment [1]. By combining photon radiotherapy (RT) with immunotherapy (IO), a local therapy can be converted into a systemic approach leading to enhanced treatment response and prolonged survival [2,3,4,5,6]. Nowadays, charged particle radiotherapy (PRT) is gaining more attention for its favorable dose-depth energy deposition profile and the capacity of heavier ions like carbons to more densely ionize, e.g., DNA, along their cell traversal [7,8,9] by higher linear energy transfer (LET). This results in formation of complex unrepairable DNA double strand breaks, thereby providing a higher relative biological effectiveness (RBE) compared to photons, in addition to a greater capacity for normal tissue sparing [10,11]. There are indications that PRT is more immunogenic than conventional photon RT, making PRT highly interesting from an IO point of view. In general, the success of RT in combination with IO is highly dependent on the following factors: (I) composition of the tumor, (II) administration of single or fractionated radiation, (III) radiation dose, (IV) radiation scheduling and (V) the type of radiation, e.g., photons or charged particles and LET [6,12,13,14,15]. These factors will be addressed in the context of conventional RT and PRT with attention to effects of radiation on the immune system and the value of immunotherapeutic approaches in combination with RT. We provide the best of knowledge on the immune-related responses triggered by PRT. More specifically, the potential of PRT towards IO advancement is discussed including the currently available prospective clinical trials of PRT and IO therapeutic combinations.

## 2. Radiation Initiates Intratumoral Immune Responses

Although the main focus of RT is based on elimination of tumor cells, the role of RT on the immune system has become of increasing interest. RT can cause intratumoral immune cells to succumb, providing a rationale for adding IO to recruit and activate immune cells [16]. Radiation can initiate immunosuppressive responses such as elevation of transforming growth factor (TGF)-β, which can stimulate naïve CD4+ T cells to differentiate into FoxP3+ regulatory T cells, suppressing effector T cell activation and proliferation [17,18]. Radiation can also increase the expression of immune checkpoint molecules associated with dampening immune responses, such as programmed cell death (PD)-1 [19]. However, RT often prevails in immune stimulation and IO can strengthen its effects considerably. For example, RT can also increase the expression levels of several pro-inflammatory cytokines, e.g., intratumoral production of interferon (IFN)-β, tumor necrosis factor (TNF)-ɑ and interleukin (IL)-1 [20,21] and hence the activation of immune cells such as dendritic cells (DCs) and B cells [22,23,24]. In turn, mainly the DCs activate T cells to become cytotoxic CD8+ T cells against cancer cells. Interestingly, the antitumor effect of RT was even abolished in type I IFN nonresponsive hosts [25]. Moreover, RT (18 Gy) was able to inhibit the proliferative, migratory and invasive capacity of cancer-associated fibroblasts, derived from patients’ non-small cell lung carcinoma (NSCLC) tissues, which are indicated as immunosuppressive stromal factors [26]. The innate and adaptive immune pathways that are initiated after conventional RT in the tumor microenvironment (TME) have been described previously in detail [27,28,29,30,31,32].

### 2.1. Conventional Radiation Induces Immunogenic Cell Death Molecules

There is a positive correlation between tumor responses to therapy and the extent of radiation-induced tumor cell death [33,34]. RT is able to induce apoptosis, primary and secondary necrosis, all leading to alterations in immunomodulatory pathways [35,36,37,38]. Cell death can lead to immunological priming in a process called immunogenic cell death (ICD), which is defined as a form of regulated cell death sufficient to activate an adaptive immune response [39]. Activation occurs through release or expression of damage-associated molecular patterns (DAMPs). DAMPS are usually not presented to the immune system, but can be found upon cytoprotective stress responses or cell death and are able to bind to pattern recognition receptors expressed by immune cells [40,41]. A multitude of DAMPs are encountered upon RT-related ICD, e.g., calreticulin translocation to the membrane, heat shock protein (HSP)-70, HSP-90 and high mobility group box (HMGB)-1 secretion and ATP passive release [42,43,44,45]. Upon single-dose RT delivery (2, 5, 10 or 20 Gy) to tumor cells, a dose-dependent increase in calreticulin and ATP could be appreciated after 24 h and for HMGB1 72 h after irradiation [46]. RT fractionation with 2 Gy during five consecutive days in glioblastoma (GBM) models led to enhanced secretion of HSP-70 and model-dependent HMGB1 secretion [47]. In patients experiencing esophageal squamous cell carcinoma (SSC) and rectal cancer, elevated HMGB1 plasma levels after chemoradiotherapy (administration weekly or on day 1 of RT, respectively) were associated with a RT enhancement effect [48,49]. A single-dose RT (18 Gy) to cancer-associated fibroblasts did not induce ICD molecules, which may indicate that tumor cells are the main source for obtaining ICD immunogenicity [50]. Hence, RT is capable of evoking an immune stimulation through cancer cell kill, although to what extent it contributes to the success of IO has not been determined.

### 2.2. Conventional Radiation Enhances (Neo)Antigen Expression

RT engages both the innate and adaptive arms of the immune system and functions as an in situ vaccine that attracts T cells and natural killer (NK) cells through DC induction. Radiation promotes the release of existing tumor antigens displayed on a tumor cell surface [51] and can even further elevate antigen expression to levels sufficient for cross-presentation, thus increasing the number of DCs presenting antigens [52]. It has been demonstrated that beside antigens, radiation can upregulate the neoantigen pool, which is attributed to the formation of novel mutations owing to error-prone radiation-induced DNA damage repair [44]. Neoantigens are highly specific per tumor type and even for each individual tumor, however low levels of neoantigen (uptake) fail in evoking a significant immune response. It was demonstrated that RT induced the expression of proteins on tumor cells that were normally not presented by major histocompatibility complex (MHC) molecules or were indicated as “silent”, including MHC-I expression [53,54], prior to radiation [55]. Similar to existing antigens, radiation-induced cell death can drive the release of existing (neo)antigens towards a detectable level for cross-presentation by DCs. It has been recently demonstrated that radiation of tumor cells (8 Gy) also enriched extracellular vesicles (EVs) with a variety of antigens and DAMPs, serving as a carrier of antigens to prime cytotoxic lymphocytes and causing significant anti-tumor response, both local and systemic, as compared to non-irradiated EVs [56]. Altogether, RT can both increase the reservoir of (neo)antigens and enhance their presentation; however, this might depend on the dose of radiation [24,57].

## 3. Dose Scheduling Effects of Conventional Radiation on Tumor Immunity

The success of IO depends highly on the radiation dose, based on the immunological variations seen systemically and within the TME after different RT treatments. Radiation sensitivity of immune cells is cell-dependent, but usually high. Sparing lymphocyte-rich organs during RT has been proposed to maintain immunotherapeutic benefits as lymphocytes are highly radiosensitive [58]. Thus, given that tumor-draining lymph nodes and circulated blood are the source of cytotoxic T cells responsible for the systemic antitumor effects, sparing these organs during irradiation may be highly favorable for patients. This is supported by a recent study, where whole body chemoradiation of patients with esophageal cancer was significantly associated with low absolute lymphocyte counts nadir, which in turn correlated with worse treatment outcome, suggesting the importance of host immunity in tumor control [59]. Detection of immune cell death in human peripheral blood following a single dose of radiation in a range of 0–60 Gy indicated that B and NK cells were most prone to cell death, followed by T cells and then the rather radioresistant monocytes and myeloid derived suppressor cells (MDSC) [60,61]. Interestingly, B and NK cells mainly died through necrosis, whereas T cells died via an apoptotic pathway, which would imply differences in immunogenic response after immune cell death. In line with these data, a recent study showed remarkable differences in the radiosensitivity between the lymphoid and the myeloid lineage decreasing in the order of T cells > NK, B cells and CD34+ progenitor cells > monocytes >> macrophages and immature DCs [62]. Peripheral blood-derived T cells (Treg, CD4+ and CD8+) started to undergo apoptosis following radiation with a dose as low as 0.125 Gy approaching to a plateau at 2 Gy with no clear threshold. Another study showed that human lymphocytes and murine macrophages had similar viability (metabolic activity) up to doses of 2 Gy, but phagocytic activity was altered [63]. As the radiation dose increased, macrophages were significantly more resistant to loss in metabolic activity than were lymphocytes.

Contrary to circulating lymphocytes, it was reported that a large proportion of intratumoral T cells survived clinically relevant radiation both fractionated (5 × 1.8 Gy) or single-dose (20 Gy), and had a similar transcriptome as radioresistant tissue resident memory T cells [64]. The irradiated surviving T cells could mediate tumor control with enhanced motility and IFN expression, without the need to recruit new T cells. These data are in line with the notion that pre-existing tumor immunity is detrimental for tumor regression and moreover demonstrated that local irradiation is not inherently immunosuppressive [65]. The presence of pretreatment matured intratumoral T cells is also important for tumor curing, because of diminished T cells priming via reduced IL-12 production by irradiated DCs [66]. Additionally, it was shown that stimulated (proliferating) T cells in peripheral blood were less radiosensitive than nonstimulated T cells, typically present in the blood in the absence of an infection, due to differences in DNA damage response (DDR) patterns induced by radiation up to the dose of 2 Gy, which could be advantageous in case of multiple dose scheduling upon T cells activation [67]. Human myeloid DCs were resistant to radiation-induced apoptosis and maintained their migratory and phagocytic capacities following radiation with a single dose as high as 30 Gy [66]. However, irradiated mature DCs were less effective at priming of naïve T cells, resulting in lower cytotoxicity against antigen-specific targets. Recruitment of DCs by RT is time-dependent as tumor infiltration occurs only between 5–10 days after initial radiation [68], thus having a need for fast antigen uptake and presentation to T cells. Immunological RT effects depend not only on the amount of dose that is given but also on whether a single dose or repetitive fractions are delivered, which has implications on immune cell recruitment. It has to be noted here that most of the studies comparing tumor response to single doses with fractionated regimes used the same total dose rather than biologically equivalent doses, which hinders the interpretation of the results. In addition, since different tumor types differ in their radiation sensitivity, it is also desirable to use isoeffective radiation doses to be able to compare responses to the combined approaches in different tumor models and to validate the findings.

### 3.1. Single Dose Schedule

What advocates for using single dose RT is that prolonged dose administration induces matrix deposition and long-term fibrotic reactions [69], apart from the evidence of reduced lymphopenia with decreasing number of fractions, for example, in breast cancer patients [70]. However, different pro- and anti-immunogenic effects can be induced depending on the level of the dose administered. Interestingly, a moderate radiation dose between 1 and 10 Gy was able to reprogram macrophage type 2 (Mɸ-2) towards a Mɸ-1 like antitumor phenotype and only a high dose of >10 Gy led to Mɸ-2 polarization [71]. High dose RT (25 Gy) drove the accumulation of CD11b+Gr-1+ neutrophils in the center of the necrotic area and CD11b-F4/80+ tumor associated macrophages at the junctions between necrotic areas and surrounding hypoxic regions [72], although the function of these cells is not fully understood yet. Furthermore, high dose RT (15–20 Gy) may permanently reduce blood flow leading to the additional formation of hypoxic immunosuppressive microenvironment, limiting infiltration of immune cells [73]. A dose of 8–16 Gy led to upregulation of acid sphingomyelinase (ASMase), which induced endothelial cell apoptosis and triggered immune responses through increased production of cytokines/chemokines that induce immune cell recruitment [74]. Moreover, a single dose of 15 Gy was more effective than consecutive 5 × 3 Gy at priming antitumor T cells to antigen ovalbumin, but priming was seen with both regimens [75]. RT dose of 6 Gy showed equal to better abscopal responses than 10 Gy and 15 Gy with an anti-CD40 agonistic antibody [76]. Low dose RT can suppress the proliferation of T regulatory (Treg) cells, specifically at a dose of 0.94 Gy more than high dose RT with 15 Gy [77]. Thus a high dose of >15 Gy appears not to be preferred in context of IO, whereas an intermediate dose range seems to provide a more balanced stimulation of pro-inflammatory and inhibition of anti-inflammatory signals.

### 3.2. Fractionation Schedule

Fractionated radiotherapy (fRT) remains the most frequently applied (curative) RT strategy in the clinic, e.g., 5 × 2 Gy per week over a 3–7 week period [78]. The immunological responses induced by fRT differ from those induced by single dose RT. For example, induction of gene expression of the TGF and IFN family was higher upon fRT delivery versus single dose RT [79]. Sustained IFN signaling, however, can contribute to the expression of the immune checkpoint programmed death-ligand (PD-L)1 and to PD-L1-independent adaptive resistance leading to tumor relapse in an in vivo study combining RT and immune checkpoint inhibitors (ICI), anti-CTLA4 and anti-PD-L1 [80]. It has also been demonstrated that fRT with 2 × 7.5 Gy resulted in similar tumor growth inhibition as 15 Gy single dose irradiation, while showing the tendency for lower Treg numbers in spleens as compared to a single dose [81]. In breast and seminoma cancer patients, fRT with 50 Gy delivered in 25 fractions in five weeks promoted the formation of memory and cytotoxic T cells whilst decreasing Tregs [82]. The seminal work of Vanpouille-Box and colleagues showed that single dosage of 12–18 Gy enhanced DNA exonuclease Trex-1 expression in different cancer cells and attenuated their immunogenicity by degrading DNA that accumulates in the cytosol upon radiation in vitro, whereas upon fRT this effect was not seen. A hypofractionated schedule of 3× 8 Gy was identified as the optimal schedule in inducing efficient local and abscopal responses in combination with ICI [83], which was subsequently used to test other immunotherapeutic agents [84]. Others also showed that a radiation dose of at least 5 Gy is required to activate inflammatory pathways, suggesting that doses below 5 Gy might not be optimal to combine with IO [85].

Recently, certain patient populations have undergone hypofractionation RT to shorten treatment duration (reduce patient treatment burden and enhance cost effectiveness) and it resulted in a similar or better toxicity profile [86,87,88]. Stereotactic ablative radiation (SABR or SBRT) delivers a higher dose per fraction in a smaller number of fractions [89,90]. It was shown that a SABR schedule of 52 Gy in 8 consecutive daily fractions (6.5 Gy per fraction) versus intensity-modulated radiation (IMRT) hypofractionated treatment of 60 Gy in 25 fractions (2.4 Gy per fraction) induced different plasma cytokine changes in NSCLC patients, with the latter schedule having a more limiting therapeutic effect [91]. SABR administered at a high dose (>7 Gy on no more than five consecutive days), increased the local production of IFN-γ, which enhances MHC-I and antigen presentation in cancer cells. Similarly, studies involving human melanoma cells confirmed that SABR with 10–25 Gy per fraction increased MHC-I expression, rendering cancer cells more susceptible to T cell attack [92]. SABR provided excellent local control rates (98% at three years and 87% at five years) in patients with NSCLC [93], but 10–20% distant metastasis and 10–15% regional lymph node recurrence was encountered [94], highlighting the need for combinatorial treatment approaches.

The type of IO adds another level of complexity in guiding the choice of radiation dose and schedule. For example, fRT, but not single-dose RT, induced an immune-mediated abscopal effect when combined with anti-CTLA-4 antibody in mouse tumor models [95]. In contrast, fRT with 5 × 2 Gy or 5 × 5 Gy combined with the immunocytokine L19–IL2 triggered only growth delay of distant tumors, while a single dose of 15 Gy resulted in complete remission of 20% of the non-irradiated tumors despite the fact that both schedules resulted in 100% cure of the primary tumor [96,97]. Similarly, fRT (3 × 4 Gy) and single dose (7 Gy) in combination with an anti-PD-L1 antibody resulted in the same therapeutic effect on the primary murine tumor CT26 [13].

## 4. Window of Opportunity to Achieve Synergy between Radiotherapy and Immunotherapy

Timing may significantly affect the generation of a long-lasting curative antitumor immune response. It should be emphasized that a RT schedule that might be seen as optimal, may be not the most optimal for the combination of RT and IO. The optimal schedule for combining radiotherapy and IO is not completely clear and may depend on the type of IO used. Namely, fRT can lead to overtreatment of patients due to the “one-size-fits-all” dose scheduling, because antitumor immunity can be mitigated after a number of fractions [98]. Moreover, dose scheduling will affect immune infiltration, which should be considered as an important factor determining IO efficacy. The optimal time window for the delivery of IO to obtain synergistic effect should coincide with the temporary radiation-induced immunogenic effects, which can also depend on RT dose and schedule. In a CT26 colorectal mouse model, immune infiltration started within a period of 2–4 days after the last irradiation with 2 × 5 Gy [68], whereas in a B16 melanoma model infiltration of CD8+ T cells could be appreciated five days after the last irradiation with 2 × 12 Gy [99]. Administration of an anti-OX40 agonist antibody was optimal when delivered one day following radiation during the post-radiation window of increased antigen presentation [100]. Our data indicate that delivery of radiation (10 Gy single dose) after the start of IO with L19–IL2 was not beneficial in a murine F9 teratocarcinoma model as opposed to the neoadjuvant schedule [97]. It also has been demonstrated that only concomitant, but not sequential administration of anti-PD-L1 antibody, i.e., seven days after completion of fractionated radiotherapy with 5 × 2 Gy, significantly enhanced overall survival of mice when compared with radiotherapy alone [101]. According to the authors, the inferior effect of the blockade of the PD1/PD-L1 signaling axis in the sequential setting was potentially due to depletion or exhaustion of tumor reactive T cells. These studies provide important information for the design of clinical trials, suggesting that IO should be combined concurrently with fRT or it should be administered shortly after ablative radiation doses, depending on the type of IO. The clinical translation of these results can be demonstrated by the recently started international multicentric randomized Phase 2 trial ImmunoSABR (NCT03705403), where L19–IL2 is combined with ablative radiation doses in a sequential manner [102].

The multifactorial dependence of the therapeutic effect of radiotherapy combined with IO highlights the need for biomarkers that would guide the selection of doses, schedules and timing to maximize the therapeutic benefit. An example of a proposed biomarker is the radiation-induced expression of DNA exonuclease Trex1 to tailor the selection of radiation dose and fractionation in patients treated with IO, in particular, immune checkpoint inhibitors [83].

## 5. Immunotherapeutic Options and Conventional Radiotherapy

Cancer cells have high adaptive skills and exploit several pathways to avoid immunological destruction. To complement RT effects, immunotherapies can enhance immunological stimuli or block evasive mechanisms. There are multiple therapies enhancing the radio-immunological response against cancer cells (Figure 1).

### 5.1. Vaccination

Vaccination strategies are often applied in relation to DCs, since antigen presentation is a crucial step in the initiation of an adaptive immunity. Typically, personalized DC vaccines, produced by isolating monocytes or hematopoietic stem and progenitor cells from peripheral blood of the patient, are subsequently treated with recombinant cytokines to induce differentiation, stimulated to induce maturation, and loaded with tumor-associated antigens (TAAs) of various forms [103]. In addition, DCs can be modulated to downregulate inhibitory molecules such as zinc-finger protein A20, which is a negative regulator of toll-like receptors (TLRs) and TNF-ɑ receptor pathways [104]. Several factors have a direct impact on DC biology and the quality and potency of the ensuing T cell responses, such as route of administration and frequency of injection, delivery system, type of adjuvants, nature of DC vaccine formulations and nature of cancer cell lysates/antigen cargo [105,106]. The difficulty in orchestrating the aforementioned factors is underlined by data showing that engineered DCs provide a shorter long-term memory effect than autologous cells [104]. However, success has already been achieved in a clinical trial, where a CD141+ subset of blood-derived myeloid DC (mDC) had superior capacities at cross-presenting TAAs to CD8+ T cells [107]. The disadvantage of vaccination is that, when applied alone, it will not overcome an immune-restrictive and suppressive TME. Herein, use of RT could empower vaccination strategies. This is supported, for example, by the results of a clinical study, where relapsed Non-Hodgkin’s B-cell lymphoma (NHL) patients benefited from pulsing DC vaccines with cancer cells dying after exposure to γ-ray, heat shock and ultraviolet-C ray associated with ICD [43]. In a Phase II clinical trial, patients with GBM received treatment with surgery, RT (60 Gy fractionated over 6 weeks) and chemotherapy followed by DC-based therapy 1–2 months post treatment initiation [108]. Increased short-term (1–3 years) survival rates were seen compared to control group receiving conventional therapy without DC vaccination. In a pilot Phase I study, six patients with diverse cancers received two four-week cycles of four intradermal daily doses of DC vaccine matured with polyinosinic-polycytidylic acid (poly-ICLC), TNF-ɑ and IFN-ɑ [109]. In addition to intratumoral injections of poly-ICLC on days 8 and 10 of each cycle and cyclophosphamide administration one week before the first cycle, SABR (3 × 8 Gy) was delivered to the selected tumor lesions between the first and the second DC vaccine cycle. DC vaccination enhanced serum IL-12 and IL-1β concentrations, confirming efficient immune stimulation. The combined treatment approach demonstrated signs of preliminary clinical efficacy. Another clinical study in patients with soft tissue sarcoma demonstrated that radiotherapy (28 × 1.8 Gy) combined with concomitant autologous intratumoral injection of DCs resulted in increased T cell responses as compared to the baseline levels prior to this therapy [110]. There are currently two active (NCT01818986, NCT01833208) and two completed clinical trials (NCT02232230, NCT01807065) in patients with prostate cancer assessing the use of RT as adjuvant in combination with sipuleucel-T. The latter is a cancer vaccine consisting of autologous antigen-presenting cells, activated ex vivo and loaded with PAP (prostatic acid phosphatase) antigen, which is expressed by the majority of prostate cancer cells, and granulocyte-macrophage colony-stimulating factor (GM-CSF) to promote DC maturation [111]. Similar to previous research, fRT is selected over single-dose radiation, which concurs with the clinical standard of care. Overall, the data suggest that combining RT with DC vaccines is a safe and promising strategy to improve patient outcomes, but requires further evidence of clinical efficacy derived from randomized clinical trials.

### 5.2. Adoptive Transfer

Currently the main research lines in adoptive transfer therapy lie in treatment with autologous in vitro expanded NK and T cells or engineered T cells. Autologous NK or T cells that have acquired antitumor activity are expanded in vitro before reinfusing the cells back into the patient [112]. T cells have been effectively isolated and expanded in treatment of naïve NSCLC tumors, even showing reactivity against tumor digests in 13 out of 17 patients [113]. Either naïve T cell or effector memory T cell populations can be used, although there is an indication that naïve T cells have superior antitumor immunity than effector memory T cells [114]. Alternatively, T cells can be engineered before infusing them into patients as either chimeric antigen receptor (CAR) T cells or T cell receptor (TCR) engineered T cells with retroviral TCRs. The difference is that retroviral TCRs can recognize intracellular and surface antigen, whereas CAR T cells can solely recognize surface antigens.

Therapy with solely adoptive transferred T or NK cells, however still encounters impairment in trafficking and persistence in suppressive solid tumors [115]. Conventional RT was able to enhance adoptive OT-I and lymph-node-derived T cell function through improved cross-priming, homing and cytotoxicity [116,117]. More specifically, it has been shown that RT induced tumor cell secretion of the cytokine IL-8 that enhanced trafficking of modified CAR T cells applying a single dose of 1 Gy in vitro and 3–4.5 Gy in vivo [118]. In addition, RT sensitized tumor cell lines MC38-OVA and EG7-OVA to the cytotoxic effects of OT-I T cells via induction of the Fas pathway. Additionally, adoptive T cells derived from splenocytes, NKG2D expressing CAR T cells, showed a synergistic therapeutic benefit in an in vitro and in vivo orthotopic GBM model when combined with RT (4 Gy) [61]. Similarly to T cells, adoptive NK cell migration and infiltration in murine tumor sites were enhanced upon RT (single-dose) and long-term survival was more pronounced as compared to NK or RT treatment alone [119]. Low dose radiation (2 Gy) sensitized heterogeneous orthotopic pancreatic tumors consisting of 25% antigen negative cells to immune rejection by locally activated CAR T cells in a TRAIL (TNF-related apoptosis-inducing ligand) dependent manner [120]. Additionally, the authors demonstrated that palliative RT (5 × 4 Gy) followed by CD19 CAR T cells in a patient with B cell lymphoma, bearing a large proportion of CD19- cells, resulted in disease-free local control one year after treatment. In another lymphoma patient study covering 11 patients, RT was used as temporary treatment for lymphoma until manufacturing of autologous T cells and CAR T cell administration had been completed. At follow-up, a complete response was attained in 5 (out of 11) patients [121]. Taken together, these studies indicate the high potential of RT not only to enhance infiltration of engineered T and NK cells, but also to increase the sensitivity of tumor cells to adoptive immunity.

### 5.3. Agonists

Immune co-stimulation through agonist receptors provides a potent signal to promote the expansion and proliferation of CD8+ cytotoxic and CD4+ helper T cells. A range of agonists is currently under investigation, such as inducible costimulatory (ICOS) agonists, TLR agonists, OX40, CD27, anti-glucocorticoid-induced TNF receptor family-related gene (GITR), 4-1BB and the cytosolic DNA sensor STING [122,123,124,125]. The GITR agonist monoclonal antibody DTA-1 has been shown to co-stimulate activated effector CD4+ and CD8+ T cells and to inhibit the suppressive activity of Tregs [126]. Also, STING agonists increase immunogenicity in non-immunogenic tumors and improve the efficacy of IO through reversing resistance to anti-PD-1 agents in mouse tumor models [127]. Agonistic agents have also been shown to significantly amplify RT-induced immune responses. It has been demonstrated in a preclinical study that fRT (3 × 8 Gy) delivered during the course of topical TLR-7 agonist imiquimod treatment resulted in significant growth inhibition of both irradiated and non-irradiated mouse breast tumors compared with either treatment [128]. This therapeutic effect was associated with increased tumor infiltration by DC and T cells. Similarly, it has been shown experimentally in a CT26 mouse colorectal cancer model that therapeutic efficacy of the small molecule TLR-7 agonist DSR-29133 was augmented by combination with fractionated irradiation (5 × 2 Gy), which was dependent on the activity of CD8+ T-cells but independent of CD4+ T-cells and NK/NKT cells [129]. Importantly, administration of the agonist either 24 h prior to, or on day 1 of fRT, but not on the last day of RT, significantly improved antitumor response. This highlights the importance of the identification of the optimal scheduling of IO and RT to achieve maximal therapeutic benefit. Another agonist, the anti-OX40 antibody, providing a co-stimulatory signal for T cell proliferation upon interaction with the OX40 receptor, has been tested in combination with RT in a mouse lung cancer model [130]. Single-dose irradiation (20 Gy) delivered on the first day of the treatment with the agonistic antibody, cured a significant proportion (50%) of the tumors, which could not be achieved with either therapy alone, which was associated with strong recruitment of antigen-specific CD8+ T cells. In line with these preclinical findings, it has been shown that single-dose RT (10 Gy) synergized with synthetic oligonucleotides (ODN) containing unmethylated cytosine-guanine (CpG) motifs which bind to TLR-9, and stimulated both innate and adaptive immunes responses, in fibrosarcoma and mammary carcinoma mouse models [131]. Another study confirmed the ability of radiation (2 × 8 Gy) to boost antitumor effects of CpG ODN in poorly immunogenic mouse breast carcinoma [132]. Combining fRT (12 × 3 Gy) with CpG ODN significantly prolonged survival of rats bearing 9L glioma, increasing the percentage of animals rejecting the tumor from ~30% after single modalities to 70% [133]. The therapeutic efficacy of this combination did not depend on scheduling, i.e., injection of CpG ODN before or after RT. The combination of low-dose fRT (2 × 2 Gy) to a tumor lesion with intratumoral injection of CpG ODN has also demonstrated the ability to induce regression of systemic disease in clinical trials [134,135]. One important limitation of the latter treatment combination is that delivery of CpG ODN relies on intratumoral injections and thus is restricted to accessible tumors. Despite the promising preclinical results exploring the therapeutic potential of RT with several co-stimulatory agents, clinical experience with this approach is limited.

### 5.4. Conjugated Cytokines

Treatment with cytokines in the past have achieved active immune responses [136], but also caused toxicities due to its unspecific targeting [137,138]. To guide cytokines selectively to the location of the target, immune-modulating cytokines can be conjugated to whole immunoglobulins (Ig), to the Fc fragment of an antibody, or fused to antigen-binding fragments specific to tumor antigens. Most cytokines are composed of a single monomeric protein domain (e.g., IL-2, IFN-ɑ) due to size advantages. Others are homodimeric (e.g., IL-10 and IFN-γ) or homotrimeric molecules (e.g., TNF, TRAIL) and some cytokines are composed of two different polypeptide chains, thus are heterodimeric (e.g., IL-12 and IL-27) [139]. The advantage of conjugated cytokines has been proven with IL-2 therapy dramatically reducing toxicity, while extending half-life from minutes to hours [140,141]. Several IL-2 conjugations have been tested, such as L19-IL2 and TNF-ɑ-IL2 [142,143]. However, even if immune cells are actively attracted to the tumor site, stimuli are still needed to target tumor cells. The combination with RT can provide this stimuli effectively [144]. The IL2-NHS immunocytokine consisting of a human NHS76 (antibody specific for necrotic DNA) fused to genetically modified human IL2, administered three days after the end of fRT (5 × 3.6 Gy), resulted in greater growth inhibition than either therapy alone in the mouse lung tumor model LLC [145], supporting the use in lung cancer patients. Interestingly, the same combination with cisplatin did not change the tumor growth but the triple combination (IL2-NHS, RT, cisplatin) resulted in the largest antitumor effect, controlling 83% of these aggressive tumors. Next, the authors tested the combination of IL2-NHS with RT (5 × 4 Gy to a single pulmonary nodule) in patients with metastatic NSCLC and showed that this approach is not only safe, but can also lead to long-term survival. IL2 was also genetically fused with a disialoganglioside D2 antibody—D2 is expressed in neuroblastoma and melanoma cells—and demonstrated complete regression of established tumors in most animals when it was combined with 12 Gy single-dose RT [146,147]. A combined therapy of ablative radiation dose (20 Gy) with a fusion protein of humanized anti-CEA (carcinoembryonic antigen) and human IL-2 (M5A-IL-2) in a transgenic murine breast and colon tumor model expressing human CEA, had significant antitumor affects as compared with RT alone [148]. It appeared that this immunocytokine generated greater therapeutic effect in combination with fRT (4 × 2.5 Gy) as compared with a single dose of 10 Gy. We have demonstrated in several preclinical studies the ability of RT to synergize with L19–IL2 and to induce durable systemic antitumor responses [96,97,149]. Our promising results prompted us to conduct clinical trials investigating the toxicity, which is deemed safe or tolerable, and therapeutic efficacy of this combination [102,150] (NCT02735850, NCT03705403, NCT04604470).

The cytokine L12 is another promising pro-inflammatory immune mediator to boost radiation-induced immune responses, which has been conjugated to tumor-specific antibodies to enhance targeted delivery, thus reducing toxicity. IL12 might be advantageous over unmodified IL2 since it might elicit TH1 responses without the risk of the expansion of Treg population unlike IL-2 [151]. fRT (5 × 3.6 Gy) combined with targeted delivery of cytokine IL12 using the NHS-IL12 immunocytokine resulted in significant growth delay compared to either treatment alone in colorectal and lung mouse tumor models [152]. In humanized mice bearing rhabdomyosarcoma xenografts, neither tumor irradiation with single dose 8 Gy, nor NHS-IL12 alone, but the combination demonstrated highly efficient systemic antitumor activity, identifying senescence and differentiation as underlying mechanisms for the observed efficacy [153]. Taken together, promising preclinical results indicate that immunocytokines in combination with RT can generate significant durable antitumor response, but clinical trials are required to proof the therapeutic benefit in clinical settings.

### 5.5. Immune Checkpoint Inhibitor Molecules and Signaling Pathways

The processes of immune cell, especially T cell, activation and suppression are tightly regulated by various cytokines as well as immune checkpoints [154]. Radiotherapy and ICI combinations have been extensively studied both in preclinical and clinical settings and showed promising results [155]. Tumor-specific T cells can upregulate expression of immune checkpoint molecules leading to T cell exhaustion; likewise, cancer cells can upregulate the expression of ligands of immune inhibitory receptors potentiating immune escape. Apart from direct induction of immune checkpoint molecules on tumor cells by RT, it has been shown that ovarian cells can express higher levels of HLA-G and PD-L1 upon mitotic arrest, indicating even more the relevance of combining RT with ICI [156]. A number of preclinical studies have shown that blockade of the PD-1/PD-L1 axis or CTLA-4 acts synergistically with RT in murine models of breast, lung, colon, skin, head and neck cancer and pleural mesothelium [101,157,158,159,160,161,162,163]. In patients with chemo-refractory metastatic NSCLC, RT (5 × 6 Gy or 3 × 9.5 Gy) and CTLA-4 blockade induced systemic antitumor T cells with 18% of patients showing objective responses and 31% disease control [164], confirming previous observations in a patient with lung cancer [165] and melanoma [166]. In Phase I clinical trials, concomitant administration of SABR and ipilimumab (anti-CTLA-4) has been well tolerated (NCT02239900), with clinical benefit associated with an increase in peripheral CD8+ T cells and the proportion of CD8+ T cells expressing 4-1BB and PD-1 [167,168]. In a prospective cohort study in patients with advanced melanoma, hypofractionated radiotherapy (26 Gy in 3–5 fractions) to one site combined with anti-PD1 therapy induced long-lasting responses with confirmed complete response in 24% of patients [168]. The significant correlation between responses in irradiated and non-irradiated areas suggested an abscopal effect. Treatment with fRT (3–5 fractions 10 Gy at 2–4 tumor sites) and pembrolizumab (anti-PD-1) against metastatic tumors showed acceptable toxicity profiles [169]. A clinical trial is ongoing to determine the effect of atezolizumab (anti-PD-L1) with hypofractionated SABR (3 fractions of 15 Gy) on metastatic tumors (NCT02992912). In a Phase 3 randomized clinical trial in patients with unrespectable stage III NSCLC (PACIFIC trial), durvalumab (anti-PD-L1) significantly improved progression-free survival as well as overall survival compared with placebo [170,171] with manageable side effects consistent with other immunotherapies. These data formed the basis for subsequent approval of durvalumab as a treatment for patients with unresectable, stage III NSCLC cancer, whose disease had not progressed following platinum-based chemoradiotherapy [172]. Moreover, in a currently ongoing Phase I/II clinical trial, patients with extracranial metastatic lesions will receive two ICIs, namely, durvalumab and tremelimumab (anti-CTLA-4) and SBRT in 3–5 fractions (total dose range 30–50 Gy) (NCT03283605) between ICI cycles. The combination of multiple ICIs with RT could provide new potential, which is supported by a recent preclinical study where combination treatment with anti-PD-1 and Indoximod, an ICI-like tryptophan, together with 2× 12 Gy RT induced rapid tumor regression [173]. This triple therapy only significantly retarded the tumor growth and led to eventual tumor relapse accompanied by increased apoptosis of intratumoral T cells. The tumors were re-irradiated (2 × 10 Gy) at a late tumor regression phase or after relapse. Only late tumor regression re-irradiation cured the majority of mice bearing melanoma or strongly delayed relapse in mice having poorly immunogenic mammary carcinoma, in agreement with more memory T cells found in the tumor draining lymph nodes and spleen.

Despite the breakthrough in treatment of cancer patients with ICIs, the overall efficiency remains unsatisfactory. More recently, malignancies, in which ICIs have shown efficacy in metastatic setting such as head and neck cancer, were investigated in combination therapies consisting of ICI and RT. However, no added benefit was found after combined treatment compared to standard RT combined with chemotherapy or targeted drugs (cetuximab) in locally advanced setting, e.g., GORTEC 2015-01 “PembroRad” or JAVELIN Head and Neck 100 trials [174,175]. Another category, where the addition of ICI has not shown any added benefit, e.g., CheckMate 548 (NCT02667587), are stroma-rich but inherently immune-cold tumors, such as GBM and pancreatic cancer. These results highlight the urgent need of novel and more successful strategies for combination therapy. New immune checkpoint molecules are being identified and explored, such as lymphocyte activation gene-3 (LAG-3), T cell immunoglobulin and mucin-domain containing-3 (TIM-3), T cell immunoglobulin and ITIM domain (TIGIT), V-domain Ig suppressor of T cell activation (VISTA) [176,177]. It has been also suggested to combine anti-PD-L1 treatment with anti-exosomal treatment to enhance antitumor response as critical evidence was found that the presence of PD-L1 in exosomes was able to exhaust T cells in lymph nodes and to reduce splenic size [178]. This exosomal PD-L1 appeared resistant to current anti-PD-L1 treatment. Upon reversal of exosome biogenesis and exosomal PD-L1 load, an effective tumor response was generated, even against at a later stage injected tumor mocking distant tumor sites. Along with the discovery of new immune checkpoint targets and the design of specific inhibitors, multiple immune modulation therapies are currently combined to enhance therapeutic outcome. Moynihan et al. described that a maximal antitumor efficacy requires four components: a tumor antigen targeting antibody, an extended half-life of recombinant IL-2, anti-PD-1, and a powerful T-cell vaccine [179]. In a currently ongoing clinical trial, GBM patients receive a personalized neoantigen vaccine together with RT and pembrolizumab (NCT02287428). In a recent study, we also combined RT with the immunocytokine L19-IL2 and the immune checkpoint inhibitor anti-PD-L1 [180]. Notably, this triple combination was most effective in the poorly immunogenic mouse lung LLC tumor model, in contrast to other two colon carcinomas, which highlights the need of biomarkers to predict which patient would benefit from this and potentially other currently tested combined approaches.

In recent years, attention has been paid to the identification and development of predictive biomarkers for the efficacy of ICIs, including biomarkers of the tumor genome and neoantigens, the tumor immune microenvironment phenotype, liquid biopsy biomarkers and host-related factors [181,182]. Once therapies are more readily available for clinical implementation, therapy selection based on biomarkers will play an important role in treatment success. This is underlined by an example of a prediction model for RT-ICI combinations, where a RT sensitivity signature based on advanced mismatch repair-deficient cancers across 12 different tumor types was overlaid to PD-L1 status. Radiosensitive patients were associated with a higher PD-L1 status compared to radioresistant patients, thus confirming the benefit from the combination [183].

## 6. Charged Particle Radiotherapy

Besides conventional RT, particle RT (PRT) such as protons (PrRT) and carbon ions (CIRT) are nowadays well-implemented therapies in clinics. Compared to conventional photons, PRT has the advantage of deeper tissue penetration, higher dose deposition at the tumor site (Bragg peak) with a steep dose gradient [184,185] and sparing more healthy tissue [186,187,188,189], thereby potentially also decreasing the irradiated blood pool (leukocytes). The relative biological effectiveness (RBE), i.e., the ratio of a reference photon or gamma radiation (6 MV photons or Co-60 γ-rays) to a dose of a test radiation, required to generate the same biological effect, of protons is considered equal to 1.1, which is currently adopted in treatment planning, whereas RBE of carbon ions is estimated to range between 2–3, depending on the tumor type, biological endpoint, etc. [190,191,192,193].

Similarly, as for conventional RT, fPRT or single-dose PRT might greatly affect the synergistic potential between PRT and IO, as already demonstrated for combined PRT and targeted therapies [194,195]. PRT fractionation thus far has shown excellent local control without severe toxicities [196,197,198]. For the radiobiology of protons and carbon ions, we refer to excellent reviews [199,200,201,202,203]. We will summarize here only the biological effects induced by charged particles that can modulate immune responses.

### Immune Modulatory Potential of Charged Particle Radiation

While there is convincing preclinical and clinical evidence on significant therapeutic benefit from combining photon RT with IO, only a limited number of comprehensive preclinical studies have investigated whether PRT elicits a different immune response compared to photons or modifies tumor cells into a more immunogenic phenotype to increase their sensitivity to immune surveillance (Figure 2).

There are indications that PRT has the potential to inflict higher immunogenicity than photons, especially CIRT. It has been reported that high-LET CIRT (>70 keV/μm) significantly increased HMGB1 levels in the culture supernatant of three human cell lines 72 h after irradiation compared with lower-LET CIRT (13 keV/μm) [204]. The comparison with photon RT was not performed in this study, but it can be assumed, based on the comparison with 13 keV/μm, that high-LET CIRT has a greater potential to enhance the release of the immunogenic molecules as compared to lower-LET RT such as linac X-rays (6–16 MeV with LET of only 0.3 keV/μm) or protons (~5 keV/µm) [205]. In another study, comparable levels of HMGB1 were found after exposure to isoeffective doses of carbon ions and photons, where HMGB1 was also measured after 72 h but with carbon ions with LET of 50 keV/μm [206]. In murine SCC, a dose-dependent release of HMGB1 and surface calreticulin expression was found after both γ-rays (^137^Cs) and CIRT, but only calreticulin expression was significantly higher after CIRT with a biologically equivalent dose [207]. A similar radiation-induced (8 Gy) upregulation of surface molecules involved in immune recognition (HLA, ICAM-1, and the tumor-associated antigens CEA and MUC-1) was seen regardless of the type of RT, in several human cancer cell lines [208]. Both photon RT and PrRT induced cytotoxic T lymphocytes lysis mediated by calreticulin cell-surface expression. In addition, PrRT increased translocation of calreticulin in both cancer stem-cell- and non-stem-cell-like populations, potentially facilitating the immune attack on cancer stem cells, which are regarded as being more radioresistant; however, this potential was not tested. Whether induction of calreticulin translocation in cancer stem cells is a unique effect observed after PrRT remains to be elucidated. Calreticulin expression was compared 48 h after photon RT, PrRT, and CIRT in four different human cancer cell lines [209]. The study concluded that proton RT and PrRT were equally effective in inducing calreticulin expression, whereas CIRT had significantly stronger effects on increasing calreticulin levels compared to PrRT and photon RT at 2 and 4 Gy. In this study, however, despite reporting different RBE values for different cell lines, comparison of calreticulin expression at isoeffective doses for each individual cell line was not performed, which affects the data interpretation. Moreover, it should also be noted that for evaluation of the immunogenic potential of a cytotoxic insult upregulation of multiple ICD molecules is required (HMGB1, calreticulin, HSPs, ATP) to trigger a solid immune response [41,210]. In a recent study, it was shown that isoeffective doses of carbon ions induced higher levels of phosphorylated mixed lineage kinase domain-like (MLKL) protein, which mediates and regulate necroptosis, providing an important source of DAMPs and ICD molecules [211].

It may be expected that PRT leads to a higher release of antigens and greater diversity of T cell repertoire (neo-antigens) than photons because clustered DNA lesions, defined as two or more individual lesions within one or two helical turns of the DNA, commonly observed with charged particles, might result in more unrepaired DNA damage and genomic mutations or instability [212,213]. DNA damage in human pancreatic cancer cells, as measured by γH2AX foci up to 24 h after irradiation with isoeffective doses, was more persistent 24 h after CIRT (50 keV/μm) as compared to 200 kVp X-rays especially in cancer cells that had a stem cell phenotype [214], which was paralleled by the larger foci size [212]. Similarly, PrRT resulted in more and persistent DNA damage in glioma stem cells, as assessed using the DNA comet assay up to 48 h after radiation exposure as compared to photon RT, supporting higher probability of antigen accumulation after PRT [212].

The concept of increased genomic instability is supported by recent studies demonstrating that PRT induces chromothripsis in cells, i.e., a form of genomic instability characterized by massive, but highly localized chromosomal rearrangements following generation of more shattered chromosome domains along the particle track in the nucleus [215]. This phenomenon leads to a new genome configuration and the formation of complex chromosomal alterations by provoking inaccurate rejoining of chromosome fragments [10]. While this event can lead to the development of secondary malignancy in the case of normal tissue exposure, it can result in the formation of neoantigens provoking stronger immune reaction, particularly when combined with IO. However, it still remains to be determined whether neoantigens induced by PRT are immunogenic. Furthermore, micronuclei, by-products of mitotic progression, formed in response to radiation, may result in the initiation of chromothripsis (the micronuclei hypothesis [216]) and can trigger activation of inflammatory signaling and are a repository for the pattern-recognition receptor cyclic GMP–AMP synthase (cGAS) [85]. CIRT and PrRT are more effective in the induction of micronuclei than photon RT, suggesting that PRT may have greater potential to activate immune response via the aforementioned mechanisms [217,218,219]. The other consequence of the complex DNA damage induced by high-LET radiation including PrRT (12 keV/μm) is specific induction of histone H2B ubiquitylation on lysine 120, which is essential for repair of clustered DNA damage. This was not observed for low-LET PrRT (1 keV/μm) and photon RT [220]. It has been shown that this process is catalyzed by E3 ubiquitin ligases MSL2 and RNF20/RNF40 complex, which modulates inflammation and inflammation-associated cancer in mice and humans. It has been shown that downregulation of RNF20 augmented the TNF-ɑ response in mouse colonocytes and innate immune cells, suggesting that induction of RNF20 would improve TNF-ɑ profile [221]. In addition, RNF20-deficient mice had more myeloid-derived suppressor cells (MDSCs) than wildtype mice, hence increased RNF20 in response to clustered DNA damage might enable an antitumor immune reaction via decreased MDSC activation. Along with the indirect immune modulating effects of PRT via induction of pro-immunogenic processes in tumor cells, CIRT (170 KeV/μm) has been shown to directly enhance phagocytic activity of murine macrophages as compared to 250 keV photon RT, while there was no difference between the effects of CIRT and photon RT on macrophage vitality [63]. Moreover, an enhanced immunogenic effect of CIRT was found to augment in vivo antibody-dependent cellular toxicity [222,223].

Numerous cytokines are involved in the regulation of radiation-induced inflammatory response by recruiting immune cells, which are crucial for both local and systemic tumor responses to radiation. The impact of PrRT on the profile of inflammatory cytokines has been investigated in several studies. Specific genes coding pro-inflammatory molecules such as IL-6, IL-8, CCL2 were less expressed in HNSCC cells following PrRT as compared to photon RT especially after fractionated irradiation with three fractions one week apart either with 2 Gy or 8 Gy per fraction [224]. In line with this result, a panel of pro-inflammatory factors was also less expressed after PrRT in fibroblasts and in blood samples of irradiated mice as compared to photon RT [225,226,227]. While lower expression levels of molecules regulating inflammation following PrRT would decrease the risk of late radiation-induced tissue injury such as fibrosis, it still remains to be determined how this would impact the immune-stimulating effects of PrRT. Whether CIRT differentially affects the expression of pro-inflammatory cytokines as compared to photon RT is yet unknown.

It has been reported and summarized in several reviews that tumor hypoxia creates an immunosuppressive (tumor) microenvironment [228,229,230,231]. Hypoxia-inducible factor 1ɑ (HIF-1ɑ) is one of the mediators of the immune escape, for example, via upregulation of immune checkpoint protein PD-L1 on dendritic cells, macrophages and tumor cells [232,233]. Although hypoxia is the main factor contributing to HIF-1ɑ stabilization, radiation-induced reactive oxygen species have been shown to stabilize HIF-1ɑ also in the presence of oxygen [234,235]. The latter effect was observed only after photon RT, whereas CIRT attenuated HIF-1ɑ signaling. This phenomenon observed for carbon ions does not only prevent adaptive prosurvival cellular processes in response to HIF-1ɑ induction, but may also result in better recognition of tumor cells by immune cells in contrast to photon radiation, supporting the potentially superior immunogenicity of CIRT. It is well established that the efficacy of PRT is less dependent on the presence of oxygen as compared to photons [236,237]. It has been shown that CIRT (5 × 1.5 Gray equivalent (GyE)) eradicated patient-derived GBM stem cells in an orthotopic xenograft tumor model, while isoeffective photon irradiation led to cancer stem cells enrichment resulting in a worse outcome as compared with CIRT [237]. Given that cancer stem cells reside in the hypoxic niche, these data suggest that immune modulatory effect of CIRT may be less dependent on hypoxia-associated immunosuppression, which is in line with the observation that the eradication of the glioma cancer stem cells correlated with the switch of transcriptome signatures related to hypoxia-associated pathways in the latter study. The same authors performed a series of experiments in murine tumor models and suggested that prolonged survival after CIRT was attributed to the reduction of Mɸ2-like macrophages and MDSCs, increased influx of CD8+ T cells, generation of an immunopermissive niche, as opposed to photon RT. Overall, it can be expected that PRT will be especially effective in combination with immunotherapies in hypoxic tumors, which requires further investigations.

The greater immunogenic potential of PRT, as compared to photons observed in some of the aforementioned in vitro experiments, has been translated in some animal studies. Mouse osteosarcoma cells irradiated with carbon ions formed less distant metastases compared to photons after exposure to isoeffective single dose of 10 Gy (5 GyE) [238]. The difference was explained by decreased metastatic capabilities of CIRT as compared to photon RT assessed by cell migration and invasion and the expression of integrins. However, it cannot be excluded that CIRT triggered immune responses protecting animals from metastatic spread, which is also supported by the greater response of primary tumors to the RBE-weighted CIRT dose. It was suggested that patients with localized prostate cancer treated with CIRT have a lower risk of subsequent primary cancers than those treated with photon RT [239]. CIRT also resulted not only in efficient elimination of the primary tumor, but also in a dramatic reduction of tumor formation after secondary tumor challenge at a contralateral site in the murine SCCVII model, indicating the development of the protective immunological memory [240]. In a different mouse SCC (NR-S1), the incidence of lung metastases decreased with increasing dose but this effect did not depend on radiation quality [241]. Clinically, a case has been reported, where a patient with an inoperable metastatic retroperitoneal sarcoma was treated with palliative PrRT and had complete regression of non-irradiated metastases consistent with the abscopal effect [242], supporting the potency of PrRT to induce systemic antitumor immune response. Taken together, preclinical and clinical data provide some evidence that PRT can trigger superior antitumor immune response than conventional photon RT, but further thorough investigations are warranted.

## 7. Charged Particle Radiation in Combination with Immunotherapy

Currently, a few preclinical studies have investigated the therapeutic effect of PRT combined with IO. In a mouse osteosarcoma model, CIRT (5.3 Gy) combined with anti-PD-L1 and anti-CTLA-4 resulted in growth delay of primary and abscopal tumors with a beneficial immune profile [243]. It is, however, unclear whether the effects obtained are equal to conventional RT, as a photon RT control arm was not included. Moreover, CIRT antitumor effects were enhanced significantly after combination with DC-based IO [244]. Similarly, the antitumor effect of CIRT was enhanced by combining it with DC-based IO, however, it remains to be determined if this combined therapy outperforms the combination with conventional RT [240]. The combination of CIRT (6 Gy) with injection of a-galactosylceramide–pulsed DCs into the primary murine SCC NR-S1 tumors effectively inhibited pulmonary metastases, but comparison with photons was also not performed [245]. In a different study using the same mouse tumor model, CIRT combined with intravenous injection of DCs resulted in significantly fewer lung metastases than photons using a biologically equivalent dose of 4 Gy (2 GyE) [207]. In a recent study using an osteosarcoma mouse model, the addition of checkpoint inhibitors (anti-PD-1 and anti-CTLA-4) to CIRT (10 Gy, 50 keV/μm) or X-rays (10 Gy) only slightly improved the response of primary tumors, the growth of which was efficiently suppressed by both radiation types upon monotherapy at least during the 21 days of the experiment [246]. The combined treatments also efficiently inhibited the growth of abscopal tumors regardless of radiation quality. In the same mice, CIRT alone reduced the number of lung metastases more efficiently than photon RT, and in combination with IO, both radiation types suppressed metastasis outgrowth, but with greater efficiency for carbon ions. This is the first comprehensive study comparing CIRT and photon RT in combination with immune checkpoint inhibitors. However, the authors used the same physical dose of carbon ions and X-rays, despite the fact that RBE values of 1.5–2 based on clonogenic assay and an RBE of 3 based on growth delay assay for this cancer cell line have been reported, which makes the interpretation of the data biased [247]. To date, results on the efficacy of protons in combination with IO have not been reported.

Clinically, using “proton therapy” as a search term in clinicaltrial.gov shows that over 600 clinical trials are registered involving PrRT, but only 8 (mainly Phase I, Table 1) are reported for the combination with specifically ICI amongst which fRT+ anti-PD1 (NCT03764787) Phase I/II, protons + anti-PD-1 (NCT03765190, NCT03087760) with unknown fractionation schedule, (hypo)fractionation with anti-PD-L1 (NCT02648997, NCT03539198, NCT03818776, NCT02444741) followed by surgical resection (NCT03267836). The search term “carbon ion therapy” retrieved 33 hits and specifically the combination with IO presented the ongoing clinical trial, where CIRT is combined with camrelizumab (anti-PD-1) for locally recurrent nasopharyngeal carcinoma (NCT04143984, Table 1). A hypofractionated CIRT trial with 40 Gy in five fractions for intrahepatic cases was initiated in combination with subcutaneous injections of GM-CSF (NCT02946138); however, enrolment was too slow and the trial was retracted. The trial could have shed light on immunotherapy combinations other than ICI with PRT. Another Phase II trial will recruit NSCLC patients for SABR CIRT with immunocytokine L19-IL2 (NCT03705403).

The centers that facilitate PrRT and CIRT are much lower in number than for conventional RT, thus, clinical trials will not be executed on a similar level. This means that trials should be designed carefully and thoroughly. The data that are available on PRT and immunity encourage an intricate, but relevant collaboration in the future of PRT and IO therapy. Since the majority of the abovementioned trials are currently in the recruiting phase, these data have to be awaited. The cost and the amount of PrRT facilities might explain the higher numbers of PrRT compared to CIRT studies; however, both PRT types are important to investigate and identify consequential responses.

## 8. Discussion and Perspectives

With the advent of immunotherapeutic agents, synergistic effects with radiation and immunotherapy have been discovered. However, obtaining a long-lasting robust immune response without tumor relapse remains difficult. PRT could open up a new era of immuno-oncology for patients who do not respond well to conventional RT. Clinical trials so far have shown that PRT has a similar, if not better, tumor control and toxicity profile as conventional RT. PRT is hypothesized to be more immunogenic than conventional RT through the upregulation of immune markers such as inflammatory cytokines or immunogenic cell death molecules and complex DNA damage, which could lead to enhanced accumulation of (neo)antigens. In addition, since PRT is less dependent on oxygen availability for its function, it has the opportunity to relieve hypoxic suppressive effects on the immune system. Functional well-designed proof-of concept experiments accompanied by mechanistic studies to understand the underlying immunological processes are urgently required to test these hypotheses. It is important to note that the functionality of immunity should be tested before adopting early conclusions based on initial results. For example, even if neoantigens are upregulated, this does not need to correlate with increased cytotoxic T cell infiltration and thus with enhanced tumor size reduction [248]. Another example is the generation of adoptive T cells, where immune checkpoint molecules are proposed as markers of tumor-specific CD8+ T cells; however, these markers are also highly upregulated in exhausted T cells. Therefore, a more elaborate selection panel may be needed for precise isolation of adoptive T cells, which yields a high T cell specific responses [249].

Similar questions as for conventional RT are relevant for PRT regarding optimal dose and radiation scheme (single-dose or fRT), including fraction size and therapeutic combinations with IO. While in preclinical studies single doses are often used in proof-of-principle experiments to avoid the influence of the confounding radiobiological factors, clinically, based on the ongoing clinical trials, hypofractionation, especially SABR, outweighs conventional fRT. However, there is no consensus on RT scheduling due to the complexity of immune regulation and the high variability amongst cancer types and intratumoral heterogeneity, but based on preclinical evidence, it appears that concomitant scheduling is likely to provide a greater outcome also for patients. If treatment schedule intensity is reduced, e.g., to a few large single doses, cost effectiveness would be enhanced, while providing sufficient immunological stimuli. Attention must be taken regarding lymphocyte sparing radiation dose delivery and the effect on tumor-resident immune cells prior to radiation. It is highly recommended to ensure experimental designs in which PRT is compared to RBE match conventional RT as this allows for equal comparison of the data obtained. Hypothetically, CIRT would outperform PrRT and conventional RT due to a higher LET, but this has yet to be established in preclinical experiments and clinical trials.

With respect to IO, selection of the type of IO for specific cancer types can also be crucial and needs more experimental data, especially in relation to dose and dose scheduling as this might alter immunological profiles that have been detected so far solely with RT. Apart from the IO mentioned here, the field is in constant development; for example, IO of viral particles and bacterial spores is ongoing [250,251]. Currently, preclinical and clinical studies investigate multimodal IO, which seems to be a promising strategy in order to tackle multiple immunological cancer hallmarks. In addition, clear reporting on dose scheduling for PRT and the effect on the immune system plus IO will help to advance this novel partnership.

Whereas enhanced normal tissue sparing and immunogenicity advocates for PRT, the main limitation is the availability of a PRT beam. Only a few centers worldwide provide it (especially for CIRT), which should strongly encourage multicenter collaborations. In this context, treatment stratification will be detrimental based on, e.g., tumor type (e.g., aggressiveness, hypoxic, immunogenic), radiosensitivity, etc. Studies are being undertaken to investigate clinical biomarkers which could possibly direct treatment planning [252,253].

## 9. Conclusions

In conclusion, there is evidence that PRT is a promising partner for IO. Comprehensive investigations are required to understand the mechanisms underlying the potentially greater immunomodulatory activity of charged particles, both local and systemic, in in vivo experiments through employing various tumor types and dose schedules. Moreover, confirmation of clinical activity is warranted.

## Figures and Tables

**Figure 1 cancers-13-01468-f001:**
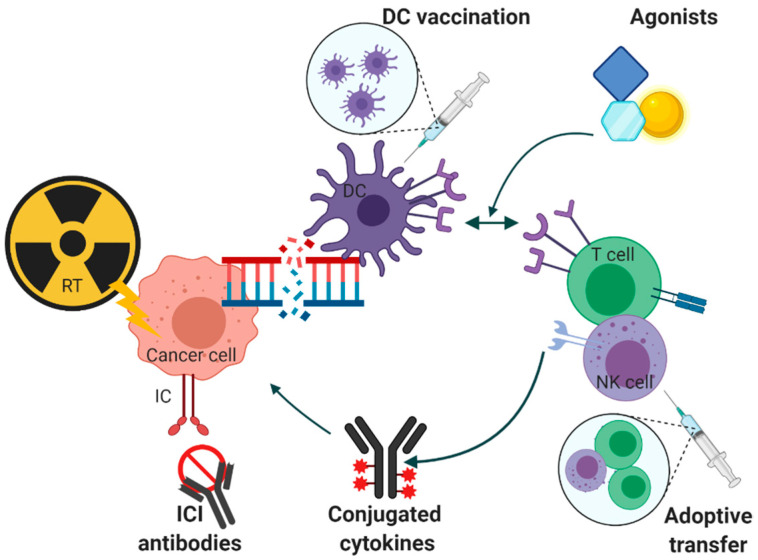
Combination of radiotherapy (RT) and different immunotherapeutic modalities. Each modality, such as dendritic cell vaccination, adoptive transfer of natural killer and T cells, agonist administration, conjugated antibodies or immune checkpoint inhibitors (ICIs), intervenes at different components of the immunological response chain. RT is able to synergize with all modalities. Created with BioRender.com.

**Figure 2 cancers-13-01468-f002:**
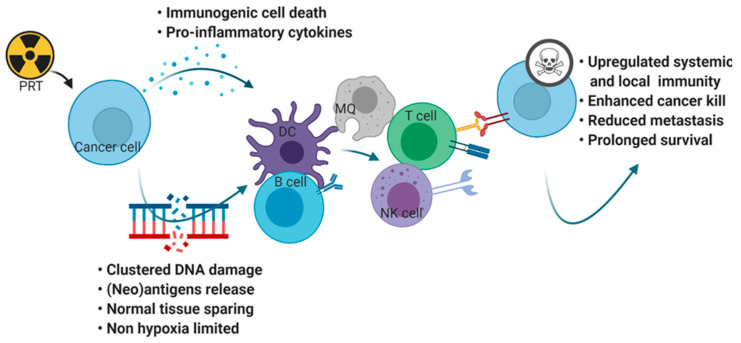
Overview of immunogenic involvement of charged particle radiation. Created with BioRender.com.

**Table 1 cancers-13-01468-t001:** Currently ongoing or initiated clinical trials regarding proton RT (PrRT) and carbon ion RT (CIRT) in combination with immunotherapy (IO).

	Identifier	Pathology	RT Dose	IO	Dose	Status	Study Type
**PrRT**	NCT02648997	Meningiomas	Unknown	Nivolumab * Ipilimumab *	N: 1 mg/kg for 3 weeks I: 3 mg/kg for 3 weeks	Recruiting	Open-label Phase-II
	NCT03267836	Meningiomas	fRT; 5 × 0.04 Gy Total 0.2 Gy	Avelumab *	Concurrent RT, 10 mg/kg, every 2 weeks for 3 months	Recruiting	Phase I
	NCT03539198	Head and neck cancer	fRT; 5× Total 35–45 Gy	Nivolumab *	Before and after RT, Q2/week for 2 weeks	Recruiting	Observational
	NCT03764787	Unknown	Unknown	a-PD-1	Unknown, for 1 year	Not yet recruiting	Phase I/II
	NCT03765190	Neoplasm metastasis	Unknown	a-PD-1	Unknown	Not yet recruiting	Phase I/II
	NCT03818776	Non-small cell lung cancer	fRT; 20–23× Total 60–69 Gy (cardiac sparing)	Durvalumab	1500 mg Q4W, max. 12 months (to 13 doses/cycles)	Recruiting	Early Phase I
	NCT03087760	Non-small cell lung cancer	Reirradiation, unknown	Pembroluzimab	Unknown	Recruiting	Phase II
	NCT02444741	Non-small cell lung cancer	fRT, 15× low dose, Total unkown	Pembroluzimab	Unknown dose for 21 days, up to 16 cycles	Recruiting	Phase I/II
**CIRT**	NCT04143984	Locally recurrent nasopharyngeal carcinoma	fRT; 21 × 3 Gy Total 63 Gy	Camrelizumab *	C: 200 mg i.v. every 2 weeks for a year maximum	Not yet recruiting	Phase II/III
**CIRT**	NCT03705403 **, [102]	Non-small cell lung cancer	SABR	Darleukin	C: 15 Mio IU, 6 cycles, 3 infusions within one cycle, every 3 weeks	Not yet recruiting	Phase II

* Nivolumab and durvalumab are PD-L1 antibodies, ipilimumab is a CTLA-4 antibody, pembroluzimab, avelumab and camrelizumab are PD-1 antibodies, darleukin is the immunocytokine L19-IL2. ** CIRT treatment arm is currently being under consideration by BfS (Federal Office for Radiation Protection, Germany). fRT: fractionated RT, Q: dose per week (Q4 is 4 doses a week), i.v.: intravenous administration.

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
