# Peer review of "Charged Particle and Conventional Radiotherapy: Current Implications as Partner for Immunotherapy"

_cancers, 2021, doi:10.3390/cancers13061468_

Round 1
Reviewer 1 Report
Comments on manuscript Marcus et. Charged particle radiotherapy: a new partner for immunotherapy?
While preclinical and clinical studies show that standard photon-based radiotherapy and immunotherapy can synergize to promote both local and systemic anti-tumor immunity, there is still room for improvement. Higher ionization density, more lethal unrepaired damage and more complex clustered DNA lesions are thought to have greater immunogenic potential and therefore other treatment modalities delivering such biological effects are being investigated. Charged particle radiation is among those and comparison is made to standard photon therapy, which is relevant. Combination of RT with immunotherapy is a popular topic with great promise. Also other RT treatment modalities, such as brachytherapy or BNCT, are proposed to be combined with immunotherapy for the very same reasons and could be briefly introduced, as has been discussed for stereotactic RT already.
I could not find any mention on exosomes in the text (except in Ref 223 Hypoxia-induced tumor exosomes promote M2-like macrophage polarization of infiltrating myeloid cells and microRNA-mediated metabolic shift). Exosomes are vesicles shed by cells and received by other cells. They are like packages that carry a range biological molecules - “messages”- and even have the address (receptors) on top of the package. They have a role as carries of information in systemic effects described in radiobiology, such as bystander or abscopal effects, as well as in promoting antitumor immunity. I think they will be very important and will have great potential as prognostic and predictive biomarkers as well as in treatment as carriers of immunological and other biological messages to/from the tumour. A couple of references on the role of exosomes in anti-tumour immunity (as example):
Lin et al. Radiation-induced small extracellular vesicles as “carriages” promote tumor antigen release and trigger antitumor immunity. Theranostics 2020; 10(11):4871-4884. doi:10.7150/thno.43539
Poggio et al. Suppression of Exosomal PD-L1 Induces Systemic Anti-tumor Immunity and Memory. Cell177, 414–427, April 4, 2019
The sentence at the end of chapter 8 Discussion and perspectives: “Studies are being undertaken to investigate clinical biomarkers [245, 246].” is not very visionary as the references start from year 1992. I suggest discussing exosomes instead.
I also came across an excellent recent paper comparing the radiosensitivity of different blood cell populations that you may find useful:
Daniel Heylmann, Viviane Ponath, Thomas Kindler, Bernd Kaina. Comparison of DNA repair and radiosensitivity of different blood cell populations. Scientific Reports (2021) 11:2478
Conclusions chapter is actually one long sentence which is a bit difficult to read and comprehend. Shorter sentences might provide clarity.
Author Response
Reviewer 1:
1) Combination of RT with immunotherapy is a popular topic with great promise. Also other RT treatment modalities, such as brachytherapy or BNCT, are proposed to be combined with immunotherapy for the very same reasons and could be briefly introduced, as has been discussed for stereotactic RT already.
Answer: we agree with the reviewer that brachytherapy or BNCT is a promising modality to be combined with immunotherapy. These were not discussed in the present review because external beam RT has been compared with charged particles in combination with immunotherapies, whereas these studies have not been reported for the former modalities. In addition, stereotactic RT is discussed because it is a modulation of conventional RT with respect to the dose per fraction, while this review discusses the immunogenic potential of external irradiation depending on the treatment scheme, i.e. fractionated vs single dose RT. We therefore feel that brachytherapy and BCNT is beyond the scope of this review.
2) I could not find any mention on exosomes in the text (except in Ref 223 Hypoxia-induced tumor exosomes promote M2-like macrophage polarization of infiltrating myeloid cells and microRNA-mediated metabolic shift). Exosomes are vesicles shed by cells and received by other cells. They are like packages that carry a range biological molecules - “messages”- and even have the address (receptors) on top of the package. They have a role as carries of information in systemic effects described in radiobiology, such as bystander or abscopal effects, as well as in promoting antitumor immunity. I think they will be very important and will have great potential as prognostic and predictive biomarkers as well as in treatment as carriers of immunological and other biological messages to/from the tumour. A couple of references on the role of exosomes in anti-tumour immunity (as example):
Lin et al. Radiation-induced small extracellular vesicles as “carriages” promote tumor antigen release and trigger antitumor immunity. Theranostics 2020; 10(11):4871-4884. doi:10.7150/thno.43539
Poggio et al. Suppression of Exosomal PD-L1 Induces Systemic Anti-tumor Immunity and Memory. Cell177, 414–427, April 4, 2019
The sentence at the end of chapter 8 Discussion and perspectives: “Studies are being undertaken to investigate clinical biomarkers [245, 246].” is not very visionary as the references start from year 1992. I suggest discussing exosomes instead.
Answer: we would like to thank the reviewer for this comment that forced us to look at determinants of tumor immunogenicity from a different angle. We added the following sentence in the respective chapter (lines 136-139): “It has been recently demonstrated that radiation of tumor cells (8 Gy) also enriched extracellular vesicles with a variety of antigens and DAMPs, serving as a carrier of antigens to prime cytotoxic lymphocytes and causing significant anti-tumor response, both local and systemic, as compared to non-irradiated EVs.” The work of Poggio is now also included in the review in the discussion of the novel targets (lines 544-552).
The references in the end of chapter 8 have been updated to make it more visionary (line 850).
I also came across an excellent recent paper comparing the radiosensitivity of different blood cell populations that you may find useful: Daniel Heylmann, Viviane Ponath, Thomas Kindler, Bernd Kaina. Comparison of DNA repair and radiosensitivity of different blood cell populations. Scientific Reports (2021) 11:2478
Answer: we thank you the reviewer for pointing our attention to this new article, which has been now included in the review (lines 158-163).
Conclusions chapter is actually one long sentence which is a bit difficult to read and comprehend. Shorter sentences might provide clarity.
Answer: Adjusted according to the suggestion.
Reviewer 2 Report
The manuscript is a review on the radiotherapy and immune therapy combination highlighting a potential treatment strategy using charged particle radiotherapy in combination with immune therapy. While the review is comprehensive and well written, several aspects require further clarifications and discussion as highlighted below.
-A clear delimitation needs to be made between the immune responses resulting from DNA damage and the immune responses induced by the tumour microenvironment. Throughout the manuscript these responses need to be discussed separately in the context of radiotherapy and charged particle therapy.
-Further clarifications are needed to highlight the use of radiotherapy and immunotherapy combinations for the different treatment strategies: local control or systemic effects. A good structure defining these outcomes, will enhance the overall clarity of the manuscript.
-If the systemic effects are mentioned, it becomes very important to discuss the radiation induced abscopal effects and the previous studies enhancing these effects using immune checkpoint inhibitors. Are there any reasons to assume these abscopal effects will be enhanced by PRT?
-Are the systemic or local PRT induced effects going to be affected by any other chemotherapy the patients will have undergone?
-Previous studies have determined a strong dose dependency of these effects, considering the amount of inflicted DNA damage. Are the authors anticipating these doses and the appropriate RBE correction to still be suitable for charged particles?
- What would be the main benefit of PRT and immune therapy combinations? Are the authors considering the improved local control or the systemic effects as the main goal of the approach? Why is that?
- What would be the authors' recommended approach considering the radiation dosing and scheduling for the proposed combination to achieve this goal?
- What is the authors opinion referring to the cost of the suggested combination therapies?
- What are potential future research avenues of immediate importance in this field?
Author Response
Reviewer 2:
The manuscript is a review on the radiotherapy and immune therapy combination highlighting a potential treatment strategy using charged particle radiotherapy in combination with immune therapy. While the review is comprehensive and well written, several aspects require further clarifications and discussion as highlighted below.
-A clear delimitation needs to be made between the immune responses resulting from DNA damage and the immune responses induced by the tumour microenvironment. Throughout the manuscript these responses need to be discussed separately in the context of radiotherapy and charged particle therapy.
Answer: We have considered this suggestion. Immune responses resulting from DNA damage or induced by the tumor microenvironment intertwined already greatly. Therefore, to prevent repetition, we decided to remain to the initial manuscript setup and discuss it jointly, whenever relevant.
-Further clarifications are needed to highlight the use of radiotherapy and immunotherapy combinations for the different treatment strategies: local control or systemic effects. A good structure defining these outcomes, will enhance the overall clarity of the manuscript. If the systemic effects are mentioned, it becomes very important to discuss the radiation induced abscopal effects and the previous studies enhancing these effects using immune checkpoint inhibitors. Are there any reasons to assume these abscopal effects will be enhanced by PRT? What would be the main benefit of PRT and immune therapy combinations? Are the authors considering the improved local control or the systemic effects as the main goal of the approach? Why is that?
Answer: We appreciate the suggestion of the reviewer to focus more on the outcomes and differentiate them (local vs. abscopal). However, originally we chose a different set-up to discuss both outcomes if available, per each type of immunotherapy combined with RT. This is also to avoid repetition because often both outcomes were investigated in one study. In addition, we stress that charged particles may have greater immunogenic potential and thus better outcomes (both local and systemic) and provide available evidence (theoretic or experimental). Unfortunately, still until now there are no solid comprehensive studies investigating the local and systemic affects of charged particles separately. Particularly, there are no reports for protons. There is only one study using CIRT and control conventional RT that investigates both outcomes but the set-up of that study is suboptimal as discussed in the review. In addition, mechanistic studies should be performed in parallel to understand the mechanisms underlying the observed effects. This is also stressed in the review that more comprehensive validation studies are required to prove that charged particles have greater immunogenic potential, which translates into the beneficial anti-tumor effects. As mentioned above, we discuss the potential greater immunogenicity of charged particles as compared with photons based on the available evidence, which can enhance both local and systemic effects but requires scientific proof and is currently being validated in different institutes including our laboratory.
-Are the systemic or local PRT induced effects going to be affected by any other chemotherapy the patients will have undergone?
Answer: The PRT induced effects might be affected by chemotherapy and any other therapy. However, we think that the topic of chemotherapy falls beyond the scope of this paper
-Previous studies have determined a strong dose dependency of these effects, considering the amount of inflicted DNA damage. Are the authors anticipating these doses and the appropriate RBE correction to still be suitable for charged particles?
Answer: Indeed, there is a dose-dependency of anti-tumor effects, which however depends on the type of immunotherapy. We think, that for the fair comparison of the effects produced by charged particles versus photons, biologically equivalent dose should be applied. In fact, we think it is not the amount of DNA damage that matters but the type of DNA damage being more complex, clustered after charged particles, which is mirrored by larger gH2AX foci. This has been discussed in the review (lines 635-645).
- What would be the authors' recommended approach considering the radiation dosing and scheduling for the proposed combination to achieve this goal?
Answer: we have discussed this in the last chapter (819-835), in particular the tendency towards hypofractionation, as well as our conclusion (literature based) on scheduling and some of the important aspects that should be considered in the design of the studies.
- What is the authors opinion referring to the cost of the suggested combination therapies?
To answer this question a cost-benefit analysis needs to be performed.
Answer: This is a very good question. In the last chapter we speculated that cost-effectiveness could be potentially enhanced if long conventional fractionated schedules can be reduced to a fewer larger fractions. This however is also true for the photon radiotherapy. To fully answer this question the comparative cost-benefit analysis should be performed, which deserves a separate publication.
- What are potential future research avenues of immediate importance in this field?
Answer: in the last chapter we summarized the still unknown components of the treatment strategies, which would result in the maximal benefit for patients. We now also stressed that “The functional well designed proof-of concepts experiments accompanied by mechanistic studies to understand the underlying immunological processes are urgently required to test these hypotheses” (lines 809-811). This is also stressed in conclusions.
Reviewer 3 Report
This manuscript described the literature review of the possibility for charged particle radiotherapy combined with immunotherapy. This is interesting topics in this field because charged particle radiotherapy is an effective local treatment in in the future and immunotherapy is expected to change systemic therapy. This manuscript is well written and summarizes the several advantages and problems, which will be useful for researchers of the charged particle radiotherapy combined with immunotherapy. So, this manuscript is worthy of being accepted in Cancers.
Author Response
No revisions were suggested.
Reviewer 4 Report
This is a well written review article in the context of the new era of immuno-radiotherapy. I would have no particular comments if the title of the paper wouldn't incline the reader towards an entirely different focus that demands a different structure of the manuscript. The current paper starts dealing with the issue stated in the title, after having reported and discussed 170 references on the effect of photon radiation on immune response. If the paper is to be published as is, the title should change to state that this is a general review on the effects of ionizing radiation (photons and particle) on immune response.
A paper with the title 'charged particle radiotherapy: a new partner for immunotherapy' demands a different approach. Authors would be called to present the effects of particle irradiation on immune response and discuss these in comparison to photon radiation. Are there indeed data supporting a differential effect? Are there other aspects of particle therapy that render these more appealing for immunotherapy combinations? These should be clearly summarized in the discussion/conclusion.
Author Response
Reviewer 4:
This is a well written review article in the context of the new era of immuno-radiotherapy. I would have no particular comments if the title of the paper wouldn't incline the reader towards an entirely different focus that demands a different structure of the manuscript. The current paper starts dealing with the issue stated in the title, after having reported and discussed 170 references on the effect of photon radiation on immune response. If the paper is to be published as is, the title should change to state that this is a general review on the effects of ionizing radiation (photons and particle) on immune response.
Answer: We agree with the reviewer and adjusted the title accordingly: Charged particle and conventional radiotherapy: current implications as partner for immunotherapy.